# Induced Pluripotent Stem Cells and Organoids in Advancing Neuropathology Research and Therapies

**DOI:** 10.3390/cells13090745

**Published:** 2024-04-25

**Authors:** Douglas Bottega Pazzin, Thales Thor Ramos Previato, João Ismael Budelon Gonçalves, Gabriele Zanirati, Fernando Antonio Costa Xavier, Jaderson Costa da Costa, Daniel Rodrigo Marinowic

**Affiliations:** 1Brain Institute of Rio Grande do Sul (BraIns), Pontifical Catholic University of Rio Grande do Sul, Porto Alegre 90610-000, Brazil; douglas.pazzin@edu.pucrs.br (D.B.P.); thales.previato@edu.pucrs.br (T.T.R.P.); joaoismaelbudelon@gmail.com (J.I.B.G.); gabriele.zanirati@pucrs.br (G.Z.); fxavier@pucrs.br (F.A.C.X.); jcc@pucrs.br (J.C.d.C.); 2Graduate Program in Pediatrics and Child Health, School of Medicine, Pontifical Catholic University of Rio Grande do Sul, Porto Alegre 90619-900, Brazil; 3Graduate Program in Biomedical Gerontology, School of Medicine, Pontifical Catholic University of Rio Grande do Sul, Porto Alegre 90619-900, Brazil

**Keywords:** iPSCs, organoids, Alzheimer’s disease, epilepsy, Parkinson’s disease, spinal cord injury

## Abstract

This review delves into the groundbreaking impact of induced pluripotent stem cells (iPSCs) and three-dimensional organoid models in propelling forward neuropathology research. With a focus on neurodegenerative diseases, neuromotor disorders, and related conditions, iPSCs provide a platform for personalized disease modeling, holding significant potential for regenerative therapy and drug discovery. The adaptability of iPSCs, along with associated methodologies, enables the generation of various types of neural cell differentiations and their integration into three-dimensional organoid models, effectively replicating complex tissue structures in vitro. Key advancements in organoid and iPSC generation protocols, alongside the careful selection of donor cell types, are emphasized as critical steps in harnessing these technologies to mitigate tumorigenic risks and other hurdles. Encouragingly, iPSCs show promising outcomes in regenerative therapies, as evidenced by their successful application in animal models.

## 1. Introduction

The investigation of neuropathologies frequently encounters challenges in establishing suitable study models. Neurodegenerative disorders like Alzheimer’s disease (AD) pose significant hurdles due to their complex interplay of extracellular and intracellular factors. For instance, understanding the immune response of patients to pathological protein aggregates like Aβ plaque, the distribution of intracellular neurofibrillary tangles (NTF), and the density of neuritic plaques proves particularly intricate [1]. In neurodevelopmental disorders, delving into the early stages of the disease poses a particular challenge when attempting to replicate it in cellular and animal models. This complexity is evident in conditions like temporal lobe epilepsies [2,3]. Conversely, when it comes to tissue injuries such as spinal cord injury (SCI) and strokes, animal models predominantly serve as the primary investigative tool, while treatment emerges as the principal obstacle [4,5]. Across all these scenarios, there exists a shared objective in exploring the potential for developing disease-specific models or therapeutic interventions.

Induced pluripotent stem cells (iPSCs) were first generated from mouse fibroblasts [6]. Subsequent studies confirmed the ability to differentiate these cells into the main neural cell types, including neurons, astrocytes, and oligodendrocytes [7]. The interest in the use of iPSCs in the field of neuropathology is due to the modeling and treatment of rare and genetically altered diseases. So iPSCs have presented the possibility for cell patient-derived disease modeling, presenting somatic mutations of interest and simulating disease phenotypes, such as AD [8]. In addition, iPSCs also present therapeutic potential in the field of regenerative therapy [9]. In this concise review, we employ a deliberate review methodology and emphasize experimental models for neuropathologies utilizing iPSCs, with a particular focus on organoid models and the prospective utilization of these cells in therapy and drug-screening pipelines.

## 2. Generation and Characterization of iPSCs

Self-renewal and pluripotency are fundamental attributes of embryonic stem cells (ESCs). Self-renewal refers to the remarkable ability to continuously proliferate without committing to a specific cell fate when cultured in vitro. Pluripotency, on the other hand, indicates the remarkable potential to differentiate into various cell types derived from the three primary embryonic germ layers [10,11]. The generation of iPSCs signifies a remarkable achievement in cellular reprogramming, providing a unique avenue to leverage the regenerative capabilities of pluripotent cells.

The formation of iPSCs was initially showcased in the groundbreaking research of Shinya Yamanaka and his team, who pinpointed a set of transcription factors capable of converting adult somatic cells into a pluripotent state. These factors, namely, Oct4, Sox2, Klf4, and c-Myc (OSKM), have since become synonymous with iPSC reprogramming [6,12]. In recent years, iPSCs have been employed across various applications, such as autologous cell therapy [13], experimental disease modeling [14,15], and as platforms for drug discovery and therapeutic screening [16]. In recent years, a wide range of protocols for iPSC generation has emerged. In the sections below, we summarize the key differences in reprogramming techniques employed for iPSC generation.

### 2.1. Which Cells Can Be Reprogrammed?

Somatic cells are specialized cells with distinct functions, such as skin cells, blood cells, or nerve cells. The concept of reprogramming posits that virtually every somatic cell in the human body harbors the potential to undergo transformation into an iPSC (Figure 1). Despite this, the efficiency and kinetics of reprogramming can vary depending on the donor cell type. Fibroblasts from both mice and humans continue to be the most prevalent cell types utilized for experimental reprogramming purposes [17]. However, in recent years, there has been a growing interest in other cell types due to their availability, therapeutic significance, and reprogramming simplicity. An example of this is CD133+ cord blood cells, which necessitate only OCT4 and SOX2 for iPSC generation [18]. Human primary keratinocytes exhibit the capability to undergo reprogramming at a rate twice as fast and with 100 times greater efficiency compared to fibroblasts [19]. Additionally, these cells offer the advantage of being obtainable through the cultivation of plucked hair from patients [20].

Urine samples also serve as a readily accessible source of cells that can be reprogrammed into iPSCs. Both renal tubular cells and exfoliated renal epithelial cells found in urine have demonstrated successful reprogramming into human iPSCs (hiPSCs) [21,22]. Functional cardiomyocytes derived from urinary cells within the cardiovascular system have displayed the ability to generate action potentials. This phenomenon has been observed both in vitro and in vivo following the differentiation of reprogrammed induced pluripotent stem cells (iPSCs) via lentiviral vector gene transduction [23,24].

Another relevant cell source for reprogramming is blood. Peripheral blood mononuclear cells (PBMCs) can be easily isolated from blood samples and stand as one of the most popular somatic cell sources for iPSC generation [25]. Both terminally differentiated mature B and T lymphocytes were able to generate iPSCs after reprogramming protocol [26,27,28], although reprogrammed T cells have been shown to induce spontaneous T cell lymphomas in mice, limiting the therapeutic applications of these cells [29]. Thus, depending on the goal, it is recommended that protocols be used to eliminate lymphocytes from the PBMCs [30]. An advantage of selecting PBMCs as the donor cells for reprogramming is that these cells can be cryopreserved and reprogrammed at a later time without compromising the kinetics and efficiency of reprogramming [31], thus enabling the utilization of frozen samples stored in blood banks worldwide [32]. It is important to highlight that several ethical, legal and social issues coexist with the availability of biological samples to researchers. Research ethics committees around the world have used policies to protect the interests of research participants, such as confidentiality, ownership, export, storage and secondary use of samples with due consent, obeying specific regulations that differ according to each country [33].

Cell type, degree of differentiation, and maturation level all exert an influence on iPSC generation efficiency. In mice, it has been documented that immature cells undergo reprogramming more readily than terminally differentiated cells [34]. Additionally, a separate study showed that hematopoietic stem and progenitor cells can yield up to 300 times more iPSC colonies compared to terminally differentiated lymphocytes [35]. Hence, selecting the appropriate cell type is a pivotal consideration prior to commencing any experiment. This decision typically hinges on cell accessibility and impacts the need for external factors, as well as the efficiency and kinetics of reprogramming.

### 2.2. The Reprogramming Recipe

The reprogramming factors, OSKM, remain the most commonly utilized method for generating iPSCs. However, in recent years, numerous alternatives have been described to refine reprogramming protocols. These include exploring other transcription factor combinations and introducing new molecules aimed at enhancing the efficiency of iPSC generation.

Given that the molecules Klf4 and c-Myc are classified as proto-oncogenes, researchers have sought substitute candidates to mitigate potential tumorigenic risks linked with these molecules. One of the initial alternative reprogramming methods was outlined by Yu and colleagues, who utilized Oct4 and Sox2 in conjunction with Nanog and Lin28. This approach resulted in a reprogramming efficiency akin to that achieved with Yamanaka’s OSKM combination [36]. An approach to reprogram mouse embryonic fibroblasts (MEFs) into iPSCs using three factors has also been developed [37]. This technique involved Oct4, Sox2, and the orphan nuclear receptor Esrrb, achieving similar efficiency compared to the OSKM protocol. Subsequent studies have further refined strategies to reprogram cells using only two transcription factors, including various combinations of Oct3/4, Sox2, Klf4, and c-Myc [38,39,40,41,42] (Figure 1).

To enhance efficiency and expedite iPSC generation, additional transcription factors have been explored for their potential to induce cellular reprogramming. Many of these factors are genes typically active during early development, playing a pivotal role in maintaining the pluripotent potential of specific cells that ultimately form the inner cell mass (ICM) in the pre-implantation embryo and contribute to embryonic development. Among the core pluripotency transcription factors are Oct4, Sox2, and Nanog. Co-expression of Nanog with the OSKM factors has been shown to halve the time required for colony appearance compared to OSKM alone [43]. Other pluripotency transcription factors, such as UTF1 and SALL4, have also been found to enhance iPSC generation when co-expressed with the OSKM factors [44,45]. Building on these findings, James Thomson devised a novel combination of transcription factors, incorporating Oct4, Sox2, Nanog, and Lin28 (OSNL), also referred to as the “Thomson factors” [36]. Additionally, Yu and colleagues explored various combinations of transcription factors, including Oct4-Sox2-Nanog-Klf4 and Oct4-Sox2-SV40LT-Klf4, to generate iPSCs [46]. These studies demonstrate that beyond the initially described OSKM factors, multiple pluripotency-related genes can be employed in different combinations to activate the pluripotency genetic program and induce cellular reprogramming into iPSCs.

Other molecules known to regulate cellular processes critical for iPSC generation and maintenance have also been employed to enhance reprogramming efficiency. Proteins that promote proliferation, such as telomerase reverse transcriptase (TERT) and the SV40 large T antigen (SV40LT), have been shown to increase colony formation when combined with OSKM [47]. Chemical compounds that positively regulate cell cycle progression, such as mitogen-activated protein kinase kinase (MAPKK) inhibitors, have also been demonstrated to increase the number of iPSC colonies obtained from reprogrammed neural precursor cells [48]. MicroRNAs (miRNAs) are also known to influence pluripotency and reprogramming [49], and several miRNAs have been tested for their capacity to increase iPSCs generation. Among these, several miRNAs from the miR-290 cluster were able to increase the number of colonies following reprogramming compared to cells using the OSKM factors alone [50]. These miRNAs are believed to be downstream effectors of c-Myc signaling but induce a population of iPSCs that is more homogeneous compared to c-Myc [51]. Numerous cells signaling pathways are regulated by miRNAs, and their potential effects on iPSCs production have been extensively reviewed elsewhere [52,53].

Encouragement for exploring safer reprogramming methods has stemmed from concerns regarding potential genetic alterations and the heightened risk of tumorigenesis associated with integrative reprogramming strategies. It has been proposed that small chemical molecules could regulate gene expression, thereby promoting the pluripotency program without inducing changes in the cell genome. The first chemically induced iPSCs were generated by replacing Oct4 with the small molecule forskolin [54]. A small-molecular combination known as “VC6T”, comprising valproic acid (an HDAC inhibitor), CHIR99021 (a GSK3 inhibitor), E-616452 (a TGF-β inhibitor), and tranylcypromine (an LSD1 inhibitor), facilitated the reprogramming of mouse cells with only Oct4 being genetically induced [55]. Recently, Guan et al. introduced another protocol for generating iPSCs solely through chemical molecules to reprogram human fibroblasts [56]. They highlighted the importance of an initial intermediate state of dedifferentiation for the small molecules to facilitate the complete reprogramming phase [56].

Another promising approach involves employing a variation in CRISPR technology to express the OSKM factors. This method utilizes a modified form of the Cas9 enzyme with inactive nuclease activity (dCas9), fused to either repressive or activating transcriptional domains. This enables CRISPR-mediated transcriptional interference (CRISPRi) or activation (CRISPRa), respectively [57]. Therefore, CRISPRa allows for the activation of gene expression in one or a specific group of genes.

Recently, CRISPRa has been successfully used to reprogram human skin fibroblasts and human leukocytes into iPSCs [58,59]. These studies have illustrated that CRISPRa represents an advanced method for iPSC generation, enabling faster and more efficient reprogramming compared to conventional protocols [58,59]. Moreover, single-cell RNA sequencing (scRNA-seq) analyses have revealed that cells reprogrammed using CRISPRa transition to the pluripotent state with high fidelity, displaying the uniform expression of pluripotency genes and minimal heterogeneity [59]. 

The versatility of the CRISPRa technique has also been described in direct reprogramming models, where mouse fibroblasts were converted into neurons through the induction of a single transcription factor [60,61]. These studies underscore that with an understanding of the genes essential for converting one cell type into another, the CRISPRa technique emerges as a reliable tool for executing this reprogramming process. This method holds the potential for achieving higher efficiency at a reduced cost and within a shorter timeframe.

Epigenetic modifications, encompassing DNA methylation and histone modifications, govern gene expression without altering the underlying DNA sequence. Throughout iPSC reprogramming, these epigenetic marks are reset to resemble those characteristic of embryonic stem cells (ESCs) [62,63]. This resetting of the epigenetic landscape is crucial for the successful conversion of differentiated cells into pluripotent stem cells [62]. Hence, the application of chemical molecules that regulate DNA methylation or chromatin modifications can enhance reprogramming in numerous cell types [64,65,66]. Treatment with histone deacetylase (HDAC) inhibitors, such as hydroxamic acid (SAHA), trichostatin A (TSA), valproic acid (VPA), and butyrate, has been shown to improve reprogramming in mouse embryonic fibroblasts (MEFs) [64,67,68]. Additionally, VPA has induced pluripotency in dermal fibroblasts and neonatal human foreskin fibroblasts (HFFs) when combined with Oct4 and Sox2 [69].

In summary, the identification of molecules that promote pluripotency and sustain stem cell states is paramount, given the relatively low success rates observed in current iPSC generation protocols. Taking into account a cell’s transcriptome and epigenetic profile is critical for selecting suitable molecules, thereby ensuring that the reprogramming process yields a sufficient number of pluripotent cell colonies.

### 2.3. Reprogramming Factor Delivery Systems

Originally, the OSKM transcription factors have been delivered into mouse and human fibroblasts using Moloney murine leukemia virus (MMLV)-derived retroviruses [70,71]. Subsequently, reprogramming was also reported using Lentivirus-based vectors [71]. Lentiviral vectors, typically derived from HIV, offer a higher cloning capacity and the ability to infect both dividing and non-dividing cells, often resulting in higher infection efficiency rates compared to MMLV-based models [72]. Additionally, Tet-inducible lentiviruses for reprogramming allow for the controlled expression of reprogramming factors [73]. However, despite achieving acceptable efficiency, their integration into the host genome has raised safety concerns. 

Since then, a variety of new delivery systems have emerged, utilizing non-integrative viral vectors such as Sendai virus and adenovirus, as well as non-viral methods including liposomes and vectors based on piggyBac transposon. [74]. (Figure 1).

General delivery systems employed in iPSCs reprogramming have been extensively reviewed elsewhere [72,73,74]. Each delivery method presents both advantages and limitations, making the selection of an appropriate delivery system an important issue to resolve before proceeding to reprogram somatic cells into iPSCs.

## 3. Organoid Models

### What They Are, How They Work, and Applications

An organoid is defined as a cell culture in a 3D structure obtained from stem cells (embryonic or reprogrammed) that after induction differentiation must consist of organ-specific cell types that self-organize. These structures are designed and function in a manner that leads to the expression of characteristics akin to various tissues and organs, including the kidneys, lungs, intestines, brain, and retina, as demonstrated in numerous studies [75,76] (Figure 2). Three-dimensional organoids are formed from human pluripotent stem cells (hPSCs), such as iPSCs and ESCs [77]. In the realm of organoids derived from iPSCs, there exists the potential to harness the combination of self-organization and differentiation capacity through genetic tools. This enables the guidance of these cells and structures toward any specific organ, a capability that could prove fundamental in the treatment of diseases [78]. In a more practical sense, organoids hold immense potential for application in advanced therapies such as organ repair through transplantable structures, as well as in drug studies. Moreover, they serve as invaluable tools for understanding the pathological mechanisms underlying certain diseases [79].

Concerning the utilization of iPSC organoids for transplantation therapies, various protocols have been developed, primarily focusing on neural differentiation to address neurological disorders like Alzheimer’s disease, Parkinson’s disease, and epilepsy (Figure 2). In this context, integrating genome-editing techniques via the CRISPR/Cas9 system can aid in modulating gene expression, thus presenting a promising therapeutic avenue for these diseases [80].

In the field of neuroscience, neural organoid models have become essential for studying various aspects of the brain, particularly neurodegenerative diseases. Their significance lies in their ability to replicate key characteristics of human brain development, which cannot be thoroughly analyzed using animal models alone [77,78]. The application of organoids enabled the formation of “mini-brains” with very specialized zones and structures such as radial glial cells and cerebral cortex, to model human microcephaly [81,82,83]. 

This methodology has progressed to the point where it is now feasible to generate highly specialized cells such as oligodendrocytes and astrocytes, and subsequently, to develop even more specialized structures in the advanced stages of neural development [84,85]. n addition to 3D organoid models, 2D models are also utilized to study brain structures, as these models facilitate the formation of neural networks [82]. However, 3D models stand as superior candidates for studying and treating neurological diseases [80].

## 4. Organoid Models in Neurological Diseases

The generation of organoids for studying neurological diseases has been presented as an important strategy for neuropathologies of different orders. Table 1 presents the main examples of neurological diseases studied from the generation of organoids.

### 4.1. Alzheimer’s Disease

Alzheimer’s disease (AD) is a neurodegenerative disorder marked by gradual memory loss, cognitive impairment, and a decline in the ability to perform daily tasks independently, ultimately becoming the leading cause of dementia [86]. The pathology of AD is characterized by the accumulation of Tau protein and the formation of amyloid beta plaques, which lead to structural changes in the brain, resulting in neuronal destruction and synaptic impairment. As a result, the molecular profile of AD is typified by the presence of amyloid beta and phosphorylated tau protein.

Another significant aspect of AD is its multifactorial nature, with a notable genetic component, where the apolipoprotein E (APOE) gene and its alleles (APOE2, APOE3, and APOE4) serve as risk factors for Alzheimer’s disease [87]. Specifically, the APOE4 allele has been strongly associated with the presence, heritability, and progression of AD [88]. Various experimental approaches, including in vivo, in silico, and in vitro methods, have been employed to comprehend and potentially treat AD. However, none have been able to fully replicate the pathological features observed in the human AD brain, despite the significant progress and promising outcomes achieved [89]. Currently, there are three commonly used methods to model AD in cerebral organoids:

Aftin-5 (Aβ42 agonist): In this model, there is an induction of APP amyloid precursor protein (Aβ) using Aftin-5 (an Aβ42 inducer that increases the production and secretion of soluble extracellular amyloid peptides). Aftin-5 treatment leads to a reproducible disruption of the physiological balance between Aβ42 and Aβ40, generating an AD-like condition in human cerebral organoids [90].

Organoids derived from familial AD (FAD): This model was established by generating iPSC-derived brain organoids from patients with familial Alzheimer’s disease (FAD) carrying APP duplications or mutations in the presenilin 1 (PSEN1) gene. Such a model can effectively recapitulate the crucial aspects of the pathology, including the presence of amyloid beta plaques and tau protein accumulation. Moreover, it provides insight into the temporal dynamics of P-tau level increases, contributing to a more comprehensive understanding of disease progression [91,92]. Another variation of this same model is the use of stem cells from patients carrying a missense mutation in the PSEN1 gene linked to early-onset AD. In this case, the cerebral organoids exhibit the same Aβ and P-tau protein aggregates as the previous model [93,94].

Model with APOE3 allele: Due to the significant association between the APOE gene and its impact on AD, organoid models with induced mutations in this gene have been developed. This model involves employing gene-editing techniques such as CRISPR/Cas9 to convert APOE3 to APOE4 in iPSCs derived from patients with sporadic Alzheimer’s disease. This is because the APOE4 variant exerts a stronger genetic influence on AD compared to other variants. In this study, it was observed that APOE4 neurons exhibited an increased number of synapses and an increased secretion of Aβ42 compared to APOE3 cells [95].

### 4.2. Amyotrophic Lateral Sclerosis

Amyotrophic lateral sclerosis (ALS) is a neurodegenerative disorder characterized by the gradual degeneration of motor neurons in the brain and the spinal cord. This degeneration leads to progressive muscle weakness, ultimately culminating in paralysis and, in many cases, death. ALS can manifest in individuals as early as their first or second decade of life, although it can also onset at later ages [96]. ALS typically starts with a localized impact and gradually spreads, resulting in symptoms that initially manifest as mild cramps or weakness in the limbs or bulbar muscles. Over time, these symptoms progress to the paralysis of nearly all skeletal muscles [97]. In terms of prevalence, the majority of ALS cases are sporadic, accounting for approximately 90% of cases, while only about 10% are hereditary, known as familial ALS (FALS). Family genes implicated in ALS are involved in various mechanisms, including proteostasis, RNA binding, and axonal transport. Specifically, genes such as SOD1, TDP-43, and PFN1 are associated with these processes, respectively [98]. One study utilized iPSC lines derived from both healthy controls and individuals affected by ALS. These iPSC lines were used to generate organoids comprising sensory neurons, motor neurons, astrocytes, and other mesodermal derivatives, including vasculature, microglia, and skeletal muscle. Interestingly, motor neurons derived from organoids of ALS-affected patients exhibited compromised Neuromotor Junctions (NMJs). Additionally, iPSC lines carrying mutations in the TARDBP, SOD1, and PFN1 genes were generated to validate the model further, confirming once more the impairment in NMJs [99].

### 4.3. Attention Deficit Hyperactivity Disorder

Attention deficit hyperactivity disorder (ADHD) is a neurodevelopmental disorder distinguished by symptoms of inattention, impulsivity, and hyperactivity. These symptoms typically manifest in early childhood and may persist even in adulthood [100,101]. ADHD compromises cognitive, behavioral, and affective domains of attention processing and executive function, response inhibition and impulsive behavior, and hyperactivity [101]. The integration of clinical, neurodevelopmental, and cognitive aspects is crucial in understanding the symptoms of ADHD. Research indicates that structural alterations in the cerebral cortex among individuals with ADHD impact cognitive processes including emotion regulation, response inhibition, and attention [102,103]. Organoid models offer a valuable approach to elucidate various aspects of neurodevelopmental disorders such as ADHD [104]. In a particular study, iPSC-derived telencephalon organoids from individuals with ADHD were utilized. Notable differences were observed between organoids derived from individuals with ADHD and those from the control group. Specifically, the ADHD group exhibited reduced cell proliferation and differentiation, along with disparities in the proportion of symmetric and asymmetric cell division. These findings suggest that the observed alterations may play crucial roles in the pathogenesis of ADHD [105].

### 4.4. Autism Spectrum Disorder

Autism spectrum disorder (ASD) represents a set of neurodevelopmental disorders with defined behavioral implications. ASD is characterized by a series of conditions, such as persistent deficits in social communication, interaction and repetitive patterns of behaviors, interests, and activities [106]. The etiopathogenesis is multifactorial with complex interactions between genetic and environmental factors [107]. ASD typically manifests in infancy or early childhood, arising from the interplay of genetic and non-genetic factors, which may act independently or in concert to contribute to its development. Over the past two decades, there has been a progressive increase in the prevalence of autism spectrum disorders (ASDs) [108]. ASD is notably more prevalent in men than in women, with data from the 2016 ADDM network indicating that ASD occurs approximately four times more frequently in men [109]. According to the DSM-5 criteria, the diagnosis of ASD requires the presence of at least two typical behavior patterns. Examples of these behaviors include repetitive motor movements, insistence on sameness or adherence to routines, echoing others’ words, hand flapping, finger snapping, fixation on specific interests, and intense attachment to particular objects [110,111]. Various factors contribute to the prevalence of ASD, including maternal factors such as advanced maternal age, obesity, diabetes mellitus, and maternal infection during pregnancy. Perinatal and neonatal factors such as umbilical cord complications, fetal distress, low birth weight or being small for gestational age, congenital malformations, and hyperbilirubinemia are also implicated. Extensive research aimed at unraveling the genetic mechanisms of ASD has identified over 800 genes associated with the condition. Additionally, hundreds of chromosomal aberrations and dozens of syndromes have been recognized. The etiology of ASD involves a complex interplay between genetic inheritance and environmental factors, which are further influenced by epigenetic processes [112]. The utilization of iPSC models and iPSC-derived organoids has been instrumental in advancing our understanding of the genetic and neurochemical mechanisms underlying ASD, as well as in exploring potential treatments for the condition. One of the pioneering studies in this field involved the generation of iPSC-derived brain organoids from individuals affected by idiopathic ASD [113]. The study revealed abnormalities in the proliferation and maturation of neurons, along with an upregulation in the expression of the transcription factor FOXG1, which correlated with the heightened production of inhibitory neurons. These findings underscore the significance of the FOXG1 gene and propose it to be a promising therapeutic target for the management of ASD [107,113]. The application of organoid iPSC models from patients with ASD made it possible to understand the mechanics and influence of genes in the etiopathogenesis of ASD. Several genes have been related to chromatin remodeling and gene transcription (MECP2, MEF2C, HDAC4, CHD8, and CTNNB1) and synaptic functions (GRIN2B, CACNA1, CACNA2D3, SCN2A, GABRA3, and GABRB3) [114]. Moreover, the utilization of cerebral organoids in ASD research has unveiled various phenotypes linked to genetic variations, impacting transcriptional pathways, morphological features, and the development of neuronal networks [115,116]. The utilization of iPSCs provides unique opportunities to investigate brain development and the ramifications of its dysregulation in neurodevelopmental disorders such as ASD. By incorporating the patient’s genetic makeup, organoid models enable the modeling of ASD while preserving the individual’s genetic background and mimicking the 3D structural complexities of the brain. Furthermore, an additional study delved into the impact of a single mutation in autism spectrum disorders (ASDs) using brain organoids. Transcriptomic analysis of these organoids, harboring a mutation in the CHD8 gene, unveiled the differential expression of key genes such as DLX6-AS1 and DLX1, which are associated with interneuronal differentiation in the CHD8 knockout organoids [117].

### 4.5. Canavan Disease

Canavan disease (CD) is a rare leukodystrophy stemming from the loss of function in the aspartoacylase (ASPA) gene within modified ASPA-expressing mature oligodendrocytes of the central nervous system (CNS) and characterized by macrocephaly, neurodevelopmental delays, and tone abnormalities [118]. The phenotypic spectrum of Canavan disease (CD) ranges from the more severe typical CD, which accounts for 90% of cases worldwide, to the less severe atypical CD, affecting the remaining 10%. Symptoms typically manifest between three and five months of age for neurodevelopmental issues in typical CD, while atypical CD symptoms become evident within the first year of life [119]. The utilization of iPSCs in the study of Canavan disease (CD) remains relatively limited, with only four studies exploring this subject for various objectives. Among these studies, two focused on engrafting mice with ASPA gene-edited iPSCs [120,121] and utilized iPSCs derived from CD subjects as a study model. Another study involved engrafting mice with wild-type ASPA gene iPSCs [122] but did not specifically target CD as a study model. Notably, only one study developed a 3D culture protocol specifically tailored for CD, a rare leukodystrophy resulting from the loss of function in the aspartoacylase (ASPA) gene, which affects altered ASPA-expressing mature oligodendrocytes of the CNS [123]. Despite the limited number of articles, all of them have demonstrated promising results in treatment, modeling, and potentially translational studies.

### 4.6. Epilepsy

Epilepsy comprises a diverse array of central nervous system (CNS) disorders characterized by an elevated susceptibility to seizures [124]. Seizures arise when neural networks are irregularly formed or disrupted by abnormal structural, infectious, or metabolic issues, resulting in anomalous firing patterns within a specific area of the brain (focal epilepsy) or across the entire brain (generalized epilepsy) [125]. Around 50 million people of all ages worldwide have their lives negatively affected by epilepsy [126].

Utilizing iPSC-derived neurons from individuals with epilepsy can provide valuable insights into the molecular and pathological mechanisms underlying certain epilepsy phenotypes [83]. Therefore, it becomes feasible to investigate neuronal behavior without relying on resected brain tissue. Incorporating editing techniques can further enhance the findings in studies utilizing iPSCs for epilepsy models [80]. 

For diseases associated with known mutations, cellular models derived from iPSCs become particularly appealing. Conversely, for epileptogenic cortical malformations or developmental epileptic encephalopathies, experimental models utilizing organoids gain prominence, as they enable the creation of intricate multi-tissue cultures.

### 4.7. Frontotemporal Dementia

Frontotemporal dementia (FTD) manifests as a neurodegenerative clinical syndrome marked by progressive deficits and alterations in behavior, executive function, motor skills, and language [127]. The World Health Organization (WHO) estimates that over 55 million individuals worldwide currently suffer from dementia, with low- and middle-income nations contributing 60% of these cases. Moreover, approximately 10 million new cases are reported annually [128]. Presently, the incidence of FTD is estimated at 1.61 to 4.1 cases per hundred thousand individuals per year [129]. These values allow us to state that frontotemporal dementia is the second most prevalent subtype of dementia, with rates ranging from 3% to 26% [130]. FTD predominantly affects individuals aged between 45 and 65, although cases have been observed in individuals under 30 years of age. While over half of FTD cases are sporadic, up to 40% of cases have a familial history associated with dementia, psychiatric disorders, or motor symptoms. Notably, at least 10% of FTD cases exhibit an autosomal dominant inheritance pattern [131]. Several genes are implicated in the manifestation of FTD symptoms. The GRN gene is accountable for approximately 60% of all cases of inherited frontotemporal lobar degeneration. Another significant gene associated with frontotemporal lobar degeneration is C9orf72, mutations in which contribute to about 25% of familial cases and represent the most prevalent genetic cause of both frontotemporal dementia and amyotrophic lateral sclerosis [132]. Several biological mechanisms that lead to frontotemporal lobar degeneration have been identified and investigated. However, the cause and many other questions remain unanswered [127]. Various studies utilizing iPSCs have contributed significantly to the understanding of disease mechanisms and potential treatments. In the context of FTD, a study employing human iPSC brain organoids expressing tau-V337M elucidated distinct mechanisms linked to MAPT mutation. This research delineated a sequence of events preceding neurodegeneration, uncovering molecular pathways associated with glutamate signaling as potential therapeutic targets for intervention in FTD [133]. Another study employing a derived brain organoid slice model successfully recapitulated the pathological conditions of C9ORF72 ALS/FTD. Within these organoids, distinct disruptions of transcription, proteostasis, and DNA repair in astroglia and neurons were observed. Consequently, patient-specific iPSC-derived cortical organoid slice cultures represent a reproducible translational platform for investigating preclinical mechanisms of ALS/FTD, as well as novel therapeutic strategies [134].

### 4.8. Huntington’s Disease

Huntington’s disease (HD) is a neurodegenerative disease caused by a CAG trinucleotide repeat increase in the huntingtin gene (HTT) [135]. The HTT mutation can lead to protein aggregation, disrupting cellular processes, particularly the basal ganglia and cortical regions of the brain [136,137]. The neuronal dysfunction ultimately leads to choreiform movements, psychiatric symptoms, dystonia, bradykinesia, and dementia [137,138].

Since HD is caused by a genetic mutation, current treatments primarily focus on managing symptoms through pharmacological interventions targeting mainly dopamine modulation [139,140]. Indeed, a more comprehensive understanding of the pathophysiology and progression of the disease is imperative to develop enhanced treatment options. Various in vitro and in vivo models have been employed to scrutinize the disease mechanisms, encompassing the introduction of the HTT mutation or the insertion of CAG repeats into the cells of both invertebrate and vertebrate animal models [141,142]. While these models have proven instrumental in unraveling the fundamental mechanisms of neuronal dysfunction and neurotransmitter dysregulation, they fall short in recapitulating the more intricate and clinical manifestations of the disease. One promising alternative has been studying the disease using cells from patients carrying the mutated gene. The scarcity of biological material can be overcome by integrating the iPSC technology into these models. In 2008, the Daley laboratory pioneered the creation of human iPSC-based models for Huntington’s disease (HD). Notably, they successfully developed the initial iPSC line from an HD patient with 72 CAG repeats. The cells were subsequently transformed into GABAergic, DARPP32-positive neurons, underscoring the potential of iPSCs to be directed into striatal neurons, a critical cell type vulnerable to degeneration in HD [143,144,145]. Subsequently, other groups developed additional iPSC cell lines from different patients [146,147,148].

Using iPSCs from patients to model HD has revealed many cellular alterations caused by the mutation. Genes related to DNA damage control pathways were downregulated in neurons derived from iPSCs (iNeurons) from patients with high CAG repeat mutations [149]. The malfunction of DNA damage repair systems can be connected to the somatic instability and mosaicism observed in HD [150]. HD-derived iNeurons showed multiple abnormalities in neuronal patterning [151,152] and an observed persistent mitotic population [153]. These changes in neuronal differentiation patterns can be linked to alterations in neurodevelopmental gene expression profiles linked to HD.

Changes in gene expression are believed to be one of the mechanisms that lead to neurodegeneration in HD. Comparisons of gene expression between iNeurons derived from HD iPSCs and gene-corrected control lines revealed the upregulation of the transforming growth factor beta (TGF-β) pathway in HD [154]. Studies conducted on iNeurons from additional HD patients further corroborate this finding [155,156,157]. Recently, the National Institutes of Health (NIH) established a consortium known as the “HD iPSC Consortium” to explore gene expression and functional changes linked to HD. RNA sequencing (RNA-seq) analysis conducted by this consortium revealed transcriptomic alterations in numerous pathways related to the development and master regulators of neurogenesis [157].

Overall, the utilization of iPSCs to comprehend the pathophysiology of HD has unveiled numerous pathways that could be targeted therapeutically to potentially mitigate the disease. Given that iPSCs can be differentiated into various cell subtypes found in brain tissue, including microglia and astrocytes, it would be intriguing to investigate whether the observed alterations in iNeurons persist in cell–cell interaction models.

### 4.9. Multiple Sclerosis

Multiple sclerosis (MS) is a chronic, inflammatory, and autoimmune disease characterized by demyelination and axonal lesions in focal and generalized regions of the brain and spinal cord. This pathological hallmark results in axonal damage to neurons in both white and gray matter areas. Clinically, MS manifests a range of characteristic symptoms that can lead to irreversible neurological disability and cognitive decline. The etiology of MS is multifactorial, influenced by complex gene–environment interactions, including genetic inheritance and lifestyle factors such as night work, excessive alcohol or caffeine consumption, and a history of infectious mononucleosis [158,159,160]. In 2020, it was estimated that there were 2.8 million people with MS worldwide. The disease is predominantly found in individuals of European descent, but it is relatively rare among individuals of Asian, African, and Native American ethnicities [161]. In terms of gender, MS is more prevalent in women, occurring twice as often as in men. By age group, MS generally affects young adults, ages 20 to 40, although some patients experience a demyelinating event during childhood or adolescence, typically with a form of multiple sclerosis called relapsing-remitting or relapsing-remitting MS (RRMS) [162,163]. The heritability of MS is polygenic, involving polymorphisms in several genes, each associated with a slight increase in the risk of developing the disease. Notably, the HLA class I and HLA class II genes confer the highest risk of MS transmission [158,164]. The use of iPSC-derived organoids is essential both to understand the mechanisms of MS and its clinical subtypes and to try to apply therapies to improve the affected areas and inhibit the effects of MS. The utilization of iPSC-derived organoids is crucial for comprehending the mechanisms underlying MS and its clinical subtypes, as well as for exploring potential therapeutic interventions to ameliorate affected areas and mitigate the effects of MS. A study involving iPSC organoids derived from both MS patients and healthy individuals revealed several key insights into MS pathology. Organoids derived from MS patients exhibited a diminished proliferation capacity compared to those from healthy controls, likely due to a reduction in the pool of stem cells. Furthermore, there was a notable decrease in SOX2+ cells and Olig2+ oligodendrocytes in MS organoids compared to controls, indicating a compromised capacity for differentiation and remyelination in MS patients and their subtypes [165]. hiPSCs can be used in the formation of oligodendrocyte progenitors generated and applied in biomedical studies. The process begins with the formation of the embryoid body, and the standardization of cells using a combination of factors T3, OLIG2, SOX10, A2B5, NG2, PDGFRα, and O4. After 3 weeks of differentiation, OPCs will form oligodendrocyte units with small subpopulations of GFAP+ astrocytes (~1%) and MAP2+ neurons (~5%) [166]. García et al. developed a human oligodendrocyte model specific to primary progressive MS. They achieved this by transducing NPCs derived from two hiPSC lines from primary progressive MS patients with SOX10 lentivirus. Their findings revealed that all lines, whether from healthy or diseased individuals, generated O4+ cells and MBP+ cells within 22 days post-transduction, exhibiting morphology and markers characteristic of intermediate and late oligodendrocytes. This organoid model offers a valuable opportunity to model MS and explore its mechanisms [167].

### 4.10. Parkinson’s Disease

Parkinson’s disease (PD) is a devastating neurodegenerative disorder characterized by the progressive loss of dopaminergic neurons in the substantia nigra region of the brain, leading to a reduction in dopamine levels and impaired fine motor coordination. Clinically, PD presents with symptoms such as bradykinesia, muscle rigidity, resting tremors, and postural instability. Pathologically, the disease is characterized by the presence of Lewy body aggregates composed of α-synuclein protein. Genetic factors play a significant role in PD, with several genes implicated in both dominant and recessive familial forms of the disease. These genes include SCNA, LRRK2, PINK1, PARK2 (parkin), and GBA1, along with others such as DJ-1, PARK9 (ATP13A2), SJ-1, and VPS35. Given the complex nature of PD, there is ongoing debate regarding potential in vitro techniques for therapeutic intervention and understanding various aspects of the disease [168].

Indeed, recent advancements in using induced pluripotent stem cells (iPSCs) and 3D brain organoid models offer promising avenues for understanding the pathogenesis of Parkinson’s disease (PD) and identifying potential therapeutic targets. By deriving neurons from patients with PD and forming organoids, researchers can analyze pathogenic mechanisms in detail and conduct drug-screening experiments. While therapeutic measures to modify the course of PD are still lacking, the use of iPSCs and organoid models holds great promise for advancing our understanding of the disease and developing novel therapeutic interventions [169]. Some organoid models are presented below:

α-synuclein (SNCA) model: α-synuclein is a protein that assumes different conformations dictated by cellular stress and is involved in neurodegenerative diseases such as PD [170]. Mutations of A53T mutant α-synuclein or α-synuclein accumulation in neurons lead to increased nitrosative stress, mitochondrial dysfunction, disrupted synaptic connectivity, transcriptional changes in synaptic signaling genes, and reduced ratio of α-synuclein tetramer to monomer, important factors in the pathogenesis of PD [171]. The iPSC-derived neuron model has triplicated levels of α-synuclein and could be a good model for understanding the morphophysiological divergences between healthy neurons and mutant neurons from PD patients.

LRRK2 model: Leucine-rich repeat kinase 2 (LRRK2) is a multikinase involved in roles in neurite outgrowth, phosphorylation of multiple proteins, and endocytic sorting via interactions with Rab-GTPases. Mutations in the gene encoding LRRK2 imply a significant risk for PD as well as other factors [172]. LRRK2 organoid models showed increased levels of oxidized dopamine and lysosomal receptor for chaperone-mediated autophagy. Also, neurons in this model have greater apoptotic activity, reducing neurite growth. Interestingly, LRRK2 also demonstrated irregularities in synaptic vesicle recycling, leading to disrupted synaptic vesicle endocytosis and decreased vesicle density in neurons.

PINK1 model: PINK1 (PTEN-induced kinase 1) is a phosphatase and tensin (PTEN) homologous protein/kinase. PINK1 localizes to the mitochondrial membrane after its depolarization, where it phosphorylates Parkin. Together, PINK1 and Parkin regulate mitochondrial health, and mutations in either related genes are associated with autosomal recessive diseases and early-onset forms of PD. A model with iPSC-derived neurons from patients expressing nonsense (Q456X) or missense (V170G) PINK1 exhibits mitochondrial defects, including impaired recruitment of Parkin to mitochondria [173].

GBA model: The GBA1 gene is responsible for encoding glucocerebrosidase (GCase) or β-glucosidase, a lysosomal enzyme that catalyzes the hydrolysis of glucosylceramide (GlcCer) into glucose and ceramide, but also the hydrolysis of D-glucosyl-N-acylsphingosine into D-glucose and N-acylsphingosine. Studies have confirmed that there is a relationship between PD and GBA mutations, including insertion, deletion, frameshift, and point mutations in GBA. Approximately 5–10% of PD patients carry GBA1 mutations. The result of the GBA mutation is the accumulation of lipids in neurons. In this PD model, alterations of GBA and GCase substrates, glycolipids glucosylceramide (GlcCer), and glucosylsphingosine (GlcSph) are found at increased levels, resulting in defective action of cellular organelles of neurons, making neurons more vulnerable to apoptosis [169].

Idiopathic Parkinson’s: The neuron-based organoid model of idiopathic Parkinson’s seeks to understand aspects of the disease that do not necessarily involve genetic aspects while also seeking to understand PD. Interestingly, it has been shown that neurons derived from patients with idiopathic PD have decreased mitochondrial respiration, increased levels of oxidized dopamine and oxidized DJ-1, and decreased GCase enzyme activity [174].

### 4.11. Spinal Cord Injury

Spinal cord injury (SCI) refers to damage inflicted upon the spinal cord, leading to a debilitating neurological and pathological condition. It profoundly impacts motor, sensory, and autonomic functions, and can also have far-reaching effects on psychological, social, and vocational aspects, significantly diminishing the patient’s overall well-being and reducing life expectancy [175].

Over the past three decades, the global prevalence of spinal cord injury (SCI) has seen a notable increase, rising from 236 to 1298 cases per million inhabitants. Additionally, it is estimated that between 250,000 and 500,000 new cases of SCI occur worldwide each year. This increase in prevalence underscores the significance of SCI as a global health concern and highlights the need for continued research and advancements in treatment and prevention strategies [176].

Initially, SCI arises from mechanical trauma (primary injury) to the spinal cord. This trauma may result from a fracture, dislocation, compression, shear, laceration, and other forms of injuries to the spine or intervertebral disc that compromise the structure of the spinal cord. Consequently, a secondary injury arises, which involves a cascade of events that are characterized by ischemia and physiological changes such as pro-apoptotic signaling and peripheral inflammation, cellular infiltration, release of pro-inflammatory cytokines and cytotoxic debris (DNA, ATP, and reactive species of oxygen) that cyclically increase the harsh post-injury situation that can result in an expansion of the lesion zone of the affected neural tissue [177].

In any case, evaluating the status of each injury resulting from SCI is essential for making therapeutic decisions. Due to the different aspects involved in the pathology of SCI, different therapeutic strategies have been proposed to treat neurodegenerative events and reduce neuronal damage resulting from SCI [178].

These efforts consist of developing neuroprotective and neuroregenerative therapies that promote neuronal recovery and improve outcomes. These therapies include spinal precautions (logrolling and cervical collar) and protection against other injuries. Additionally, there are treatment options, both surgical and pharmacological, although the efficacy of some of these treatments is still under debate. The pharmacological treatments employed often target alpha-adrenergic and beta-adrenergic functions with the aim of reducing the effects of SCI by controlling pressure and interfering with the mechanisms of action of inflammatory cytokines [176,177].

Although significant progress has been made in the understanding and treatment of LM, a definitive cure has yet to be made a reality due to the complexity of the pathology. However, new therapies based on induced pluripotent stem cells (iPSCs) and organoid models are currently, being studied to promote the treatment of patients affected by SCI and/or other neuromotor diseases. Cell-based treatments include transplants of autologous Schwann cells, olfactory ensheathing cells, mesenchymal stem cells, neural precursor cells, oligodendrocyte progenitor cells, and macrophages to the affected area [175].

Furthermore, iPSC can be used to construct 3D organoid models customized according to the aspects of each disease studied. In the case of SCI, some components studied are part of the structures affected by SCI [179].

Among the organoid models studied are early neuroectodermal cells generated from astrocytes and programmed through the overexpression of OCT4 and p53, along with the supply of molecules such as CHIR99021, SB431542, RepSox, and Y27632. Direct reprogramming of astrocytes into neurons may pave the way for in vivo neural organogenesis from endogenous astrocytes for the repair of central nervous system injuries [180]. 

Another study generated a three-dimensional culture system and protocol for the production of human spinal cord-like organoids (hSCOs), recapitulating the neurulation-like morphogenesis of the early spinal cord. hSCOs exhibited neurulation-like tube-forming morphogenesis, cellular differentiation into the major types of spinal cord neurons as well as glial cells, and mature synaptic functional activities, among other features of spinal cord development. In this study, the hSCOs generated were used to screen for antiepileptic drugs that can cause neural tube defects. hSCOs may also facilitate the study of human spinal cord development and the modeling of diseases associated with neural tube defects [181].

Organoids modeling the ventral spinal cord have been successfully generated, exhibiting the capacity to differentiate into various constituent cell types, including motor neurons, excitatory V2a interneurons, inhibitory Renshaw interneurons, and astrocytes. The presence of this heterogeneous cellular composition within the organoid framework provides a platform for elucidating the complex pathological mechanisms underlying spinal cord injuries (SCIs). Emerging evidence underscores the significance of cell-to-cell interactions within the three-dimensional microenvironment, emphasizing their pivotal role in deciphering the cellular intricacies governing SCI pathology [182].

### 4.12. Stroke

Stroke, a cerebrovascular disease resulting from a disruption of blood flow to the brain, stands as a primary cause of neurological impairment and mortality among the elderly population globally. Despite advancements in risk factor management and therapeutic interventions, there persists a concerning rise in the incidence of stroke cases [183]. Since the inception of iPSCs for stroke research, the primary focus has been on cell engraftment, driven by the promising history of stem cell engraftment in ameliorating stroke outcomes. Animal models typically employed include the rodent middle cerebral artery occlusion model for ischemic stroke and intracranial hemorrhagic models for hemorrhagic stroke. While iPSC engraftment in stroke rodent models is hampered by tumorigenicity concerns, both models have exhibited favorable outcomes, including neurological protection and reduced inflammatory responses [184]. Stroke is a tough disease to model in vitro due to the involvement of its tissue heterogeneity. The most common in vitro models for stroke entail oxygen and glucose deprivation. The use of iPSCs has already been reported in this 2D model [185]. Another in vitro model for studying stroke that uses iPSCs involves modeling cerebral autosomal dominant arteriopathy with subcortical infarcts and leukoencephalopathy (CADASIL), which leads to vessel weakening due to the accumulation of the NOTCH3 protein. This condition is associated with mutations in the NOTCH3 gene, resulting in the gain or loss of cysteine residues in specific exons. iPSCs have been successfully utilized to model this disease, offering insights into its pathogenesis and potential therapeutic interventions [186,187,188]. Given this context, the utilization of organoids derived from iPSCs has emerged as an attractive model for simulating vessel weakening and neurological damage characteristic of stroke conditions in a 3D in vitro setting [189,190].

### 4.13. Traumatic Brain Injury

Traumatic brain injury (TBI) is defined as a severe injury to the head resulting from a forceful impact. Its clinical manifestations are diverse and multifaceted, often encompassing edema, traumatic axonal injury, contusion, and hematoma. TBI is a leading cause of mortality and disability, affecting an estimated 69 million individuals worldwide [191]. TBI poses significant challenges for both in vitro and in vivo modeling. In vivo models often struggle to accurately replicate the human response to injury due to differences between human and mouse brains. On the other hand, in vitro models may fall short in capturing the complex interplay of processes that occur throughout the entire body [192]. TBI is divided into two major injuries: The primary injury (PI) occurs in the form of the impact and structural damage simultaneously, causing axonal shear, cell death, and inflammation, enabling the most common clinical signs and even death [193]. Multiple processes that occur during an extended period and include a cascade of metabolic, neuroinflammatory response, and degenerative changes [194], which may lead to several neurodegenerative diseases, including AD, chronic traumatic encephalopathy (CTE), PD, and other forms of dementia or movement disorders characterize the secondary injury (SI) [195,196,197]. The utilization of iPSCs in TBI research primarily centers around potential regenerative therapies and in vitro modeling studies. Various research groups have successfully developed TBI-related assays employing the iPSC technology. The majority of these studies concentrate on specific modeling targets, such as the primary impact (PI) effects [198,199], SI neuroinflammatory response [200], and PI traumatic axonal injury [201].

A protocol developed by Santiago-Ramirez et al. in 2021 introduced a potential approach using 3D organoids to evaluate the effects of TBI. These organoids, derived from iPSCs reprogrammed from fibroblasts, exhibited a variety of cells and structures found in the brain and displayed behavior akin to the cerebral cortex. However, despite the promising results, this model still lacks certain cell types and structural components needed to fully replicate the spectrum of the TBI pathology [202].

Another strategy to study TBI is based on iPSCs (NS/PS), which, through genome editing, expressed the yeast cytosine deaminase/uracil phosphoribosyl transferase enzyme-prodrug gene (yCD-UPRT)These NS/PS exhibited high expression of yCD-UPRT. which was observed in in vivo bioluminescent imaging and histopathological analyses in mice models of TBI. The result obtained was that the group treated with NS/PS showed improvement when compared to the control group. Furthermore, significant functional improvements in motor skills and prevention of brain atrophy were observed in mice treated with NS/PCs. Prevention of brain atrophy was observed in mice transplanted with NS/PCs [192].

In a particular study, iPSCs were derived from reactive glial cells extracted from the adult neocortex affected by TBI. The iPSC generation process involved the use of retroviruses and four transcription factors, i.e., Oct4, Sox2, Klf4, and c-Myc, in an in situ approach, where iPSCs were induced within the tissue affected by TBI. These iPSCs subsequently differentiated into abundant neural stem cells, which further matured into various cell types, including neurons and glia. Remarkably, effective brain repair was observed, characterized by the presence of numerous neurons exhibiting typical neuronal morphology, complete with axons, dendrites, and the ability to generate action potentials [203].

**Table 1 cells-13-00745-t001:** Organoid models generated for studying several neuropathological diseases.

Organoid Type	Disease	Cell Type	Result	Reference
Cerebral Organoid	AD	iPSC	Modeling sporadic Alzheimer’s disease in human brain organoids under serum exposure	[204]
Cerebral Organoid	AD	hiPSC	Mechanisms of hyperexcitability in Alzheimer’s disease hiPSC-derived neurons and cerebral organoids vs. isogenic controls	[205]
Cerebral Organoid	AD	iPSC	Modeling amyloid beta and tau pathology in human cerebral organoids	[206]
Disease Stem Cell	AD	iPSC	Familial Alzheimer’s disease mutations in PSEN1 lead to premature human stem cell neurogenesis	[207]
Disease Stem Cell	AD	iPSC and hiPSC	iPSC-derived human microglia-like cells to study neurological diseases	[208]
Cerebral Organoid	AD	iPSC	APOE4 exacerbates synapse loss and neurodegeneration in Alzheimer’s disease patients’ iPSC-derived cerebral organoids	[209]
Cerebral Organoid	AD	iPSC	A logical network-based drug-screening platform for Alzheimer’s disease representing pathological features of human brain organoids	[210]
Cerebral Organoid	AD	iPSC	Loss of function of the mitochondrial peptidase PITRM1 induces proteotoxic stress and Alzheimer’s disease-like pathology in human cerebral organoids	[211]
Cerebral Organoid	AD	iPSC	Tau pathology epigenetically remodels the neuron-glial cross-talk in Alzheimer’s disease	[212]
Disease Stem Cell	AD	iPSC	APOE4 causes widespread molecular and cellular alterations associated with Alzheimer’s disease phenotypes in human iPSC-derived brain cell types	[95]
Disease Stem Cell	AD	iPSC	Type I interferon signaling drives microglial dysfunction and senescence in human iPSC models of Down syndrome and Alzheimer’s disease	[213]
Cerebral Organoid	AD	iPSC	Acetylation changes tau interactome to degrade tau in Alzheimer’s disease animal and organoid models	[214]
Cerebral Organoid	PD	hiPSC	Modeling G2019S-LRRK2 sporadic Parkinson’s disease in 3D midbrain organoids	[215]
Cerebral Organoid	PD	hiPSC	Lewy body-like pathology and loss of dopaminergic neurons in midbrain organoids derived from familial Parkinson’s disease patient	[216]
Midbrain Organoid	PD	hiPSC	Human iPSC-derived midbrain organoids functionally integrate into striatum circuits and restore motor function in a mouse model of Parkinson’s disease	[217]
Neurospheres	PD	hiPSC and iPSC	Patient-derived three-dimensional cortical neurospheres to model Parkinson’s disease	[218]
Midbrain Organoid	PD	hiPSC and iPSC	Neurodevelopmental defects and neurodegenerative phenotypes in human brain organoids carrying Parkinson’s disease linked DNAJC6 mutations	[219]
Midbrain Organoid	PD	iPSC	Microglia integration into human midbrain organoids leads to increased neuronal maturation and functionality	[220]
Cerebral Organoid	PD	iPSC	Use of 3D organoids as a model to study idiopathic form of Parkinson’s disease	[221]
Cerebral Organoid	PD	iPSC	The Parkinson’s disease-associated mutation LRRK2-G2019S alters dopaminergic differentiation dynamics via NR2F1	[222]
Cerebral Organoid	Rett syndrome	hiPSC	Identification of neural oscillations and epileptiform changes in human brain organoids	[223]
Cerebral Organoid	TLE	iPSC	Modeling genetic epileptic encephalopathies using brain organoids	[224]
Cerebral Organoid	TSC	hiPSC	Amplification of human interneuron progenitors promotes brain tumors and neurological defects	[225]
Motor neurons study	ALS	iPSC	Aberrant axon branching via Fos-B dysregulation in FUS-ALS motor neurons	[226]
Sensorimotor organoids	ALS	iPSC	Human sensorimotor organoids derived from healthy and amyotrophic lateral sclerosis stem cells form neuromuscular junctions	[99]
Cerebral Organoid	ALS	iPSC	Spinal cord extracts of amyotrophic lateral sclerosis spread TDP-43 pathology in cerebral organoids	[227]
Motor neurons and brain organoids	ALS and FTD	iPSC	CRISPR/Cas9-mediated excision of ALS/FTD-causing hexanucleotide repeat expansion in C9ORF72 rescues major disease mechanisms in vivo and in vitro	[228]
Cerebral organoid slice model	ALS and FTD	iPSC	Human ALS/FTD brain organoid slice cultures display distinct early astrocyte and targetable neuronal pathology	[134]
Brain organoids	ALS and FTD	iPSC	Granulin loss of function in human mature brain organoids implicates astrocytes in TDP-43 pathology	[229]
Motor neurons	ALS	hiPSC	Exploring motor neuron diseases using iPSC platforms	[230]
Cerebral organoids	FTD	iPSC	ELAVL4, splicing, and glutamatergic dysfunction precede neuron loss in MAPT mutation cerebral organoids	[133]
Molecular study	FTD	iPSC	Pathological progression induced by the frontotemporal dementia-associated R406W tau mutation in patient-derived iPSCs	[231]
iPSC-derived astrocytes	MS	iPSC	iPSC-derived reactive astrocytes from patients with multiple sclerosis protect cocultured neurons in inflammatory conditions	[232]
Model study	MS	iPSC	Selective PDE4 subtype inhibition provides new opportunities to intervene in neuroinflammatory versus myelin-damaging hallmarks of multiple sclerosis	[233]
RRMS and PPMS iPSC cellular models	MS	iPSC	Generation of RRMS- and PPMS-specific iPSCs as a platform for modeling multiple sclerosis	[234]
Cerebral organoids	MS	iPSC	Cerebral organoids in primary progressive multiple sclerosis reveal stem cell and oligodendrocyte differentiation defect	[165]
Model study	MS	iPSC	Generation and characterization of four multiple sclerosis iPSC lines from a single family	[235]
Cerebral organoids	ASD	iPSC	Single-cell brain organoid screening identifies developmental defects in autism	[236]
Forebrain organoids/Molecular study	ASD	iPSC	Cortical overgrowth in a preclinical forebrain organoid model of CNTNAP2-associated autism spectrum disorder	[237]
Organoids/Molecular study	ASD	iPSC	FOXG1-dependent dysregulation of GABA/glutamate neuron differentiation in autism spectrum disorders	[113]
Brain organoids	ASD	iPSC	Superoxide dismutase isozymes in cerebral organoids from autism spectrum disorder patients	[238]
Organoids/Molecular study	ASD	iPSC	CRISPR/Cas9-mediated heterozygous knockout of the autism gene CHD8 and characterization of its transcriptional networks in cerebral organoids derived from iPSC cells	[239]
Cell Therapy	TBI	Rat	Combining enriched environment and induced pluripotent stem cell therapy results in improved cognitive and motor function following traumatic brain injury	[240]
Cell Therapy	TBI	Mice	Controlled cortical impact model of mouse brain injury with therapeutic transplantation of human induced pluripotent stem cell-derived neural cells	[241]
Cerebral Organoid	TBI	hiPSC	Modeling traumatic brain injury in human cerebral organoids	[198]
Cell Therapy	CD	Mice	Cell-based therapy for Canavan disease using human iPSC-derived NPCs and OPCs	[122]
Cerebral Organoid	Stroke	hiPSC	Gene expression profiles of human cerebral organoids identify PPAR pathway and PKM2 as key markers for oxygen glucose deprivation and reoxygenation	[242]
iPSC derived telencephalon organoids	ADHD	iPSC	Telencephalon organoids derived from an individual with ADHD show altered neurodevelopment of early cortical layer structure	[105]
Model study	ASD and ADHD	iPSC	Modeling human cerebellar development in vitro in 2D structure	[243]
Molecular study	ADHD	iPSC	Generation of a human induced pluripotent stem cell (iPSC) line from a 51-year-old female with attention-deficit/hyperactivity disorder (ADHD) carrying a duplication of SLC2A3	[244]
Model study	ADHD	iPSC	Generation of four iPSC lines from peripheral blood mononuclear cells (PBMCs) of an attention-deficit/hyperactivity disorder (ADHD) individual and a healthy sibling in a Caucasian family in Australia	[245]
Model study	ADHD	iPSCs and NSCs	Growth rates of human induced pluripotent stem cells and neural stem cells from attention-deficit/hyperactivity disorder patients: a preliminary study	[246]
Molecular study	HD	iPSC	An alternative splicing modulator decreases mutant HTT and improves the molecular fingerprint in Huntington’s disease patient neurons	[247]
Molecular study	HD	iPSC-derived neurons (Mice)	CryoET reveals organelle phenotypes in Huntington’s disease patient iPSC-derived and mouse primary neurons	[248]
Model study	HD	iPSC-derived neural cells	Bioenergetic deficits in Huntington’s disease iPSC-derived neural cells and rescue with glycolytic metabolites	[249]
Model study	HD	iPSC-derived neural cells	Extracellular vesicles improve GABAergic transmission in Huntington’s disease iPSC-derived neurons	[250]

Legend: AD—Alzheimer’s disease; ADHD—attention-deficit/hyperactivity disorder; ALS—amyotrophic lateral sclerosis; ASD—autism spectrum disorder; CD—Canavan disease; HD—Huntington’s disease; FTD—frontotemporal dementia; MS—multiple sclerosis; PD—Parkinson’s disease; TLE—temporal lobe epilepsy; TSC—tuberous sclerosis complex.

### 4.14. Limitations in the Use of Organoid Models

While organoid models offer numerous advantages, it is important to acknowledge their limitations, particularly in the context of disease treatment approaches. One critical consideration is their efficacy, as they may not fully replicate the complex microenvironments found in native tissues and organs. This can pose a significant limitation when translating findings from organoid studies into effective treatments [79].

iPSC-derived organoids present several overarching deficiencies. These encompass issues such as inadequate reproducibility due to variations in culture conditions yielding inconsistent experimental outcomes. Additionally, achieving a precise cell type composition akin to that of native tissues or organs poses challenges, resulting in size and shape heterogeneity. Moreover, organoids often lack crucial components including vasculature, immune cells, neural innervation, and specific morphological attributes, limiting their ability to effectively recapitulate physiological tissue environments. Furthermore, functional capabilities are frequently absent or limited in organoids, posing obstacles for studying complex biological processes and disease mechanisms. Based on these points, it has become a current objective to improve organoid protocols to both reduce these in vitro and organ effects and to promote better adaptation and expression of organ characteristics [78].

## 5. iPSC-Based Therapies for Neurological Diseases

It is likely possible to model iPSCs into all somatic cell types [251]. Indeed, investigations have pivoted towards leveraging this potential to model specific diseases at the cellular level, particularly those driven by genetic mutations. Patient-derived iPSCs have emerged as a focal point for conducting high-throughput screens aimed at elucidating the developmental mechanisms underlying various pathologies [252]. Therefore, iPSCs have been widely used in research strategies for pharmacological screening and regenerative therapy [253].

The objective of employing stem cell treatment strategies for regenerative therapy is to achieve the remission of pathological changes present in affected tissue, along with the potential to restore lost function resulting from localized damage [254]. The proliferative and differentiating capabilities of this cell type offer the potential for tissue regeneration in conditions requiring surgical resection or tissue loss, such as spinal cord injury (SCI) [255]. iPSCs in the context of regenerative therapy become valuable due to the patient-derived cells, which prevent tissue rejection, in addition to promoting recovery of function [256].

Currently, numerous studies are investigating the regenerative potential of iPSCs for neuropathologies. Animal model studies focused on spinal cord injury (SCI) have shown promising results, with high success rates in motor function recovery [257,258,259]. There have been groups that have attempted to prevent loss of function in animal models of stroke and ischemia [260,261]. In temporal lobe epilepsy, it was carried out to graft modified iPSCs to non-epileptogenic GABAergic neurons [262,263,264]. Even the use of iPSCs has become the target of study as a potential treatment of diseases by replacing tissue in which surgical intervention would not occur, such as the case of PD, metachromatic leukodystrophy (MLD), and HD [265,266,267]. Within all these pathologies, iPSCs have great regenerative and replacement potential for damaged tissue due to the possibility of editing cells to overcome the alterations resulting from the pathology [254].

Pharmacological tests conducted in vitro using various human cell models often face challenges in accurately mimicking the pathology. However, iPSCs offer a valuable opportunity to generate patient-specific cellular models, enabling the investigation of cellular mechanisms underlying diseases [268]. Although iPSCs are less commonly utilized in drug screening, they possess valuable properties for research purposes. Patient-derived cells obtained through iPSCs are highly specific, and the process of acquiring them is less invasive, making them excellent candidates for in vitro study models [269].

The modeling of diseases using iPSCs has seen a significant increase in interest, driven by the need for more precise investigations into pathologies. In diseases such as Alzheimer’s disease (AD), where the pathology is well understood, established drugs have been tested using iPSC-based models to evaluate their efficacy [270]. Currently, the direct use of new drugs in iPSC-based models is uncommon. However, strategies involving the use of iPSCs as a means to validate drug efficacy are increasingly being utilized in cellular models (Table 2) [271].

iPSCs are still under investigation to address their major challenges. While their pluripotent nature is advantageous for forming teratomas, which helps confirm their identity in early-stage studies, their tumorigenic potential is a significant concern for cell therapy. Studies have reported the formation of gliomas in the brain in graft therapy investigations [271]. Since the inception of iPSC research, numerous groups have identified significant risks associated with these cells, particularly concerning their tumorigenic potential linked to transgenic C-MYC expression and viral integration. Furthermore, the Single Cell Transcriptomics methodology has been bolstering disease modeling efforts, enabling a deeper understanding of genetic mechanisms such as gene expression. It is also instrumental in characterizing cell subtypes and identifying novel drug candidates. For instance, Fernandes et al. employed single-cell transcriptome analysis to identify and characterize cellular heterogeneity in human iPSC-derived dopamine neurons. Their study revealed distinct dopaminergic neuron subtypes within iPSC cultures, particularly after exposure to cytotoxic effects and genetic stressors induced by the drug felodipine [272]. Indeed, another study harnessed the Single Cell Transcriptomics methodology to cultivate organoids with precise developmental features and cell type characteristics of their target organ. This technique enabled the differentiation and selection of cell subgroups representing the desired organ characteristics while excluding cells that did not align with those features [273,274].

Hence, ongoing studies aimed at utilizing iPSCs for engraftment purposes are delving deep into examining alterations within the generated cellular model to preempt any undesired modifications [275,276]. Despite the challenges identified in the application of the technique, well-established groups have expressed keen interest in utilizing human pluripotent stem cells (hPSCs) for therapeutic purposes in humans [277].

**Table 2 cells-13-00745-t002:** iPSC usage in cell therapy and drug screening for several neuropathologies.

Trial Type	Disease	Target	Result	Reference
Cell Therapy	AD	Rat	The transplanted rats rescued Alzheimer’s cognition.	[278]
Cell Therapy	AD	Mouse	Grafted mice showed improved memory, synaptic plasticity, and reduced AD brain pathology, including a reduction in amyloid and tangle deposits.	[279]
Drug Screening	AD	hiPSC	β-secretase inhibitor IV (BSI) and γ-secretase inhibitor XXI/compound E (GSI) showed similar effects as screening in other models.	[280]
Drug Screening	AD	hiPSC	Docosahexaenoic acid (DHA) treatment alleviated the stress responses in the AD neural cells.	[270]
Drug Screening	AD	hiPSC	The anthelminthic avermectins increase the relative production of short forms of Aβ and reduce the relative production of longer Aβ fragments in human cortical neurons.	[281]
Cell Therapy	HD	Mice	iPSCs survived and differentiated into region-specific neurons in both mice groups without tumor formation.	[282]
Cell Therapy	HD	Mice	Grafted mice showed a significant increase in lifespan. In iPSC groups, animals showed significant improvement in motor functions and grip strength.	[283]
Cell Therapy	HD	Rat	Grafted rats showed significant behavioral improvements for up to 12 weeks. iPSCs enhanced endogenous neurogenesis and reconstituted the damaged neuronal connections.	[166]
Cell Therapy	HD	Mice	Improved neuronal dysfunction by SUPT4H1-edited iPSC grafts.	[284]
Cell Therapy	MLD	Mice	Transplantation of ARSA-overexpressing precursors into ARSA-deficient mice significantly reduced sulfatide storage up to 300 µm from grafted cells.	[285]
Cell Therapy	MLD	Mice	Grafts of iPSCs into neonatal and adult immunodeficient MLD mice stably restored arylsulfatase A (ARSA) activity in the whole CNS and a significant decrease in sulfatide storage when ARSA-overexpressing cells were used.	[165]
Cell Therapy	PD	Rat	iPSC graft differentiated into mature mDA neurons that survive over long term and restored motor function.	[286]
Cell Therapy	PD	Mice	hiPSCs differentiated into mDA neurons and long-term motor functional recovery was achieved after transplantation.	[287]
Cell Therapy	PD	Rat	Grafted iPSCs could survive in Parkinsonian rat brains for at least 150 days, and many of them differentiated into tyrosine hydroxylase (TH)-positive cells.	[288]
Cell Therapy	PD	Rat	Intranigral engraftment to the ventral midbrain demonstrated that mDA progenitors cryopreserved on day 17, and cells had a greater capacity than immature mDA neuron cells to innervate over long distances to forebrain structures.	[289]
Cell Therapy	PD	Rat	hiPSC-derived dopaminergic progenitor cells integrate better into the striatum of neonates than older rats.	[290]
Cell Therapy	PD	Mice	More than 90% of the engrafted cells differentiated into the lineage of mDA neurons, and approximately 15% developed into mature mDA neurons without tumor formation.	[291]
Cell Therapy	PD	Rat	There was a neural remodel of basal ganglia circuitry and no tumorigenicity.	[292]
Cell Therapy	PD	Mice	iPSCs matured into mDA neurons, reverse motor function, and established bidirectional connections with natural brain target regions without tumor formation.	[217]
Cell Therapy	SCI	Rat	Transplanted cells displayed robust integration properties, including synapse formation and myelination by host.	[293]
Cell Therapy	SCI	Mice	Due to DREADD expression, it was shown a significant decrease in locomotor dysfunction in SCI-grafted mice, which was exclusively observed following the neurons’ maturation.	[294]
Cell Therapy	SCI	Mice	The combination of iPSC graft and rehabilitative training therapy significantly improved motor functions.	[295]
Cell Therapy	SCI	Rat	Neuro-pluripotent cells derived from iPSC were able to survive and differentiate into both neurons and astrocytes, which improved forelimb locomotor function.	[296]
Cell Therapy	Stroke	Mice	Combination of electroacupuncture and iPSC-derived extracellular vesicle treatment ameliorated neurological impairments and reduced the infarct volume and neuronal apoptosis in MCAO mice.	[297]
Cell Therapy	Stroke	Pig	Tanshinone IIA nanoparticles increased iPSC engraftment, enhanced cellular and tissue recovery, and improved neurological function in a translational pig stroke model.	[298]
Cell Therapy	Stroke	Rat	Increased glucose metabolism and neurofunctional in iPSC-transplanted rats.	[299]
Cell Therapy	Stroke	Rat	Graft of iPSCs inhibited microglial activation and expression of proinflammatory cytokines and suppressed oxidative stress and neuronal death in the cerebral cortex at the ischemic border zone.	[300]
Cell Therapy	Stroke	Mice	Graft survived well and primarily differentiated into GABAergic interneurons and significantly restored the sensorimotor deficits of stroke mice for a long time.	[301]
Cell Therapy	Stroke	Rat	Generated oligodendrocytes survived and formed myelin-ensheathing human axons in the host tissue after grafting onto adult human cortical organotypic cultures.	[302]
Cell Therapy	TLE	Mice	A much-reduced frequency of spontaneous recurrent seizures in grafted animals.	[262]

Legend: AD—Alzheimer’s disease; HD—Huntington’s disease; MLD—metachromatic leukodystrophy; PD—Parkinson’s disease; SCI—spinal cord injury; TLE—temporal lobe epilepsy; mDA—midbrain dopaminergic. SCI and PD papers presented are only from 2023 and 2022 due to the large number of publications.

## 6. Conclusions and Future Perspectives

The utilization of induced pluripotent stem cells (iPSCs) in the investigation of neuropathologies offers a promising avenue for comprehending complex diseases and exploring potential therapeutic interventions. The advent of iPSCs represents a significant advancement in cellular reprogramming, enabling the creation of patient-specific models that effectively recapitulate the characteristics of diverse neurological disorders [303].

The versatility of induced pluripotent stem cells (iPSCs) in differentiating into key neural cell types, including neurons, astrocytes, and oligodendrocytes, has greatly facilitated the development of patient-derived disease models for various conditions such as Alzheimer’s disease (AD), Parkinson’s disease (PD), epilepsy, spinal cord injury (SCI), stroke, traumatic brain injury (TBI), Canavan disease (CD), autism spectrum disorder (ASD), attention deficit hyperactivity disorder (ADHD), and multiple sclerosis (MS). Moreover, the ability to manipulate iPSCs using gene-editing techniques has further expanded their utility, allowing researchers to study specific disease-related mutations [168,304].

The importance of selecting appropriate donor cell types and optimizing reprogramming protocols cannot be overstated. Continuous efforts are essential to improve efficiency and safety in induced pluripotent stem cell (iPSC) generation. While concerns about their applicability persist, particularly regarding tumorigenesis, several studies using animal models have reported positive results without tumor formation. Despite the controversy surrounding the use of iPSCs as therapies, ongoing research is shedding light on their potential benefits (Table 2).

The integration of induced pluripotent stem cells (iPSCs) into organoid models marks a significant advancement, enabling the in vitro recreation of 3D tissues. Derived from iPSCs, organoids offer a unique platform for studying complex structures like the brain and have yielded insights into the pathogenesis of neurodegenerative diseases such as Alzheimer’s disease (AD) and Parkinson’s disease (PD). Through the detailed exploration of organoid models tailored to each neurological condition, including specific genetic mutations and treatment approaches, this technology demonstrates immense potential in disease-modeling and drug-screening endeavors.

While induced pluripotent stem cells (iPSCs) offer tremendous potential for regenerative therapies, there are inherent challenges and limitations that must be addressed. Enhancements in organoid protocols are necessary to improve reproducibility, specificity, and functional characteristics. Moreover, the careful consideration of the tumorigenic potential of iPSCs underscores the importance of rigorous investigations and implementation of safety measures in their application for cell therapy [305,306]. Issues related to iPSC study models include their specificity, which is advantageous for mutation-specific diseases but challenging for sporadic, multifactorial, and epigenetic-dependent diseases such as Alzheimer’s disease (AD), Parkinson’s disease (PD), and amyotrophic lateral sclerosis (ALS). In rodent models of iPSCs used for drug screening, the translational efficacy of developed drugs has been limited, highlighting the value of using human iPSCs (hiPSCs). However, using hiPSCs generated from patient-specific cells for drug screening across diseases with diverse origins may compromise the effectiveness of therapy. Therefore, there is a need to discover new drug targets and standardize phenotypes for diseases like AD. Additionally, understanding the lack of translational potential for a particular treatment across multiple patients is crucial for effectively treating individual patients modeled with iPSCs [307]. Therefore, having a data bank for each modeled iPSC is of major importance.

In summary, the extensive investigation into iPSCs and their role in modeling neurological diseases, developing organoids, and exploring potential therapies offers a comprehensive view of the current research landscape in this field. The ongoing endeavors to improve techniques and overcome limitations highlight a dedication to advancing iPSC-based methodologies for gaining deeper insights into neuropathologies and fostering the development of innovative treatments.

## 7. Future Perspectives

Regenerative medicine seeks to repair and rejuvenate impaired organs or tissues, aiming to bring back their original form and function. Currently, clinical trials are being conducted to explore cell therapies aimed at replenishing vital cell types lost due to various diseases. For instance, in Parkinson’s disease, efforts are focused on replacing neurons [308]. In vivo reprogramming is increasingly seen as an appealing alternative to address the technical challenges associated with the iPSC technology, such as ex vivo reprogramming and large-scale expansion.

The process of in vivo reprogramming takes place amidst a distinctive cellular and extracellular milieu rich in tissue-specific biochemical and mechanical cues. Under these circumstances, cells are generated displaying a greater level of maturation compared to those reprogrammed in vitro [309]. For it to be successful, the identification of appropriate cell sources is crucial. Ideally, these initial cells should be abundant and permissive to reprogramming. Two types of macroglial cells, namely, astrocytes and oligodendrocyte progenitor cells (also referred to as NG2 glia), have been extensively investigated as prime candidates for conversion into induced neurons [310]. NG2 glia possess the ability to self-renew and exhibit high proliferation rates, characteristics that may mitigate the risk of depleting the native population crucial for maintaining tissue homeostasis [311]. For the insights gained from these animal studies to be applicable in clinical settings, it is imperative to attain robust reprogramming within diseased organs safely. Additionally, comprehensive regulatory protocols are required to unify and regulate the endeavors of both academic and industry sectors effectively.

## Figures and Tables

**Figure 1 cells-13-00745-f001:**
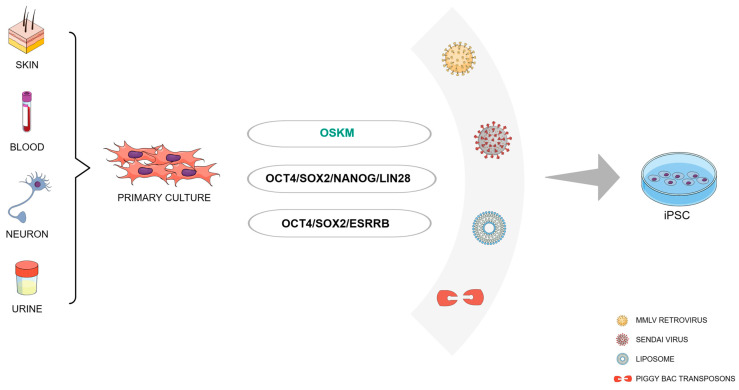
Primary cultures can be derived from various tissues including skin, blood, neurons, and urine. Cellular reprogramming for iPSC generation can be conducted using different combinations of transcription factors, with the Yamanaka factors (OSKM) remaining the most commonly employed. Similarly, there are diverse tools available to facilitate cell transfection for reprogramming, such as MMLV, SENDAI virus, liposome, and transposons.

**Figure 2 cells-13-00745-f002:**
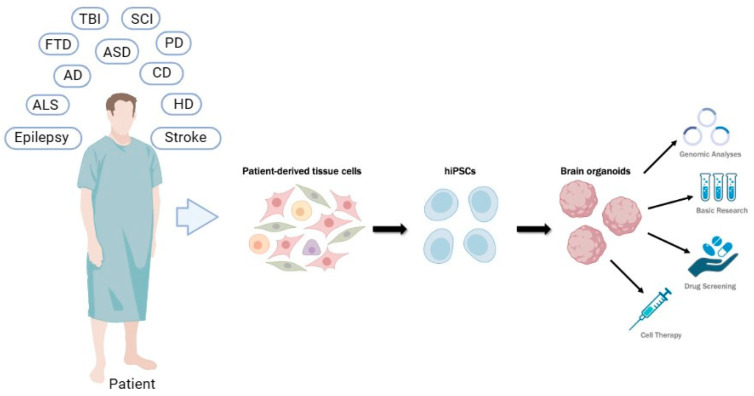
Organoids can be produced from iPSC cultures using three-dimensional cultivation techniques. Derived from iPSC cultures, these organoids can mimic various organ-specific tissues, serving as invaluable models for studying a wide range of diseases. Alzheimer’s disease (AD), amyotrophic lateral sclerosis (ALS), autism spectrum disorder (ASD), Canavan disease (CD), epilepsy, frontotemporal dementia (FTD), Huntington’s Disease (HD), Parkinson’s disease (PD), spinal cord injury (SCI), stroke, and traumatic brain injury (TBI).

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
