# Peer review of "Induced Pluripotent Stem Cells and Organoids in Advancing Neuropathology Research and Therapies"

_cells, 2024, doi:10.3390/cells13090745_

Round 1

Reviewer 1 Report

Comments and Suggestions for Authors

This is indeed an interesting and timely review on iPSCs and Organoids in Neuropathology research. The generation of iPSCs is no longer limited to OSKM reprogramming factors. Therefore authors must include newer strategies for the generation of iPSCs and organoids including small molecules, newer transcription factors, small molecules microRNAs etc. Please provide advantages and disadvantages for the original and newer reprogramming strategies. Most recently direct in vivo reprogramming strategies have been successfully developed and must be included. In addition to AD, PD, HD and SCI, authors must include Stroke, Traumatic brain Injury, Multiple Sclerosis, ALS, Canavan Disease, FTD, Autism and ADHD patient derived iPS cells for disease modeling and development of therapies. Please do a thorough literature search and revise the reference list. For example include DOI: https://doi.orh/10.1016/j.stem.2022.01.007. Additional figures with disease-specific signaling mechanisms and treatment strategies showing therapeutic targets would be necessary. 

Author Response

We are grateful for your comments and guidance to improve our article. We made the necessary corrections that can be seen in the attached file.

Reviewer 2 Report

Comments and Suggestions for Authors

In "iPSCs and Organoids in Advancing Neuropathology Research 2
and Therapies", Douglas Bottega Pazzin et al review the literature about hiPSCS, and its use in the research of therapies for neurological diseases.

This work is quite interesting, and the topic is very relevant nowadays, there are some issues with the manuscript in the present form

Methods: The approach to do this review is not clear. The authors do not indicate that the review is systematic, nor explain why some of the works are included or not. Although the expertise of some of the authors support a more freely crafted manuscript, the lack of a systematic approach causes it to cause some deficiencies in certain topics.

Organization: In some parts the work seems unorganized. Some examples include

·         focusing in different diseases in different parts of the manuscript. Although this can be caused by several reasons, it implies that the conclusions might be useful in some parts, but not all

·         the abbreviation of iPSC is repeated several times along the manuscript

·         A big part of the manuscript is focused on hiPSCs and its derivation, and later on organoids, but the approach of both parts is quite different, not dealing with these two topics in a similar, consistent way.

·         Huntington’s disease is seldom mentioned until section 3.2.3, and afterwards is not mentioned again? Other diseases are considered in every other part of the review.

Sometimes the sentences lack specificity, and this is problematic for a review.

·         In line 28, neurodegenerative diseases “present large extracellular and intracellular factors”. The nature of such factors is not clarified in this paragraph, and no further information is provided, making the significance of this sentence questionable

·         Organoids are defined as  3D “in vitro structures of cells under specific conditions.”, but it is not known what the authors mean by “specific conditions”

·         It is also not clear the importance of the sentence in line 35, “Induced pluripotent stem cells (iPSCs) were first generated from mouse fibroblasts”. The relevance for this review is not clear, might be eliminated and it would have resulted in a more readable document. Morevoer, it is not explained what are these iPSCs, and only later they explain what are the ESC and concepts such as stem cell and pluripotency, continuing with the topic of iPSC afterwards.

Repetition of similar words, such as

“focused” and “focusing” in line 43,

“important” in lines 377 and 379

Application in line 439

Use of words seems a bit strange sometimes, such as:

Oct4, Sox2, Klf4, and c-Myc are “hallmark of iPSC reprogramming”? (line 57) What does that mean?

Who can be reprogrammed? (line 63) Is it Who the correct pronoum?

Lymphocytes gave rise to iPSCs? (line 86)

Despite cell type (line 94)

Differentiate MEFS into iPSCs? (line 112)

3D organoids are formed from hPSCS, such as iPSCs, “but can be formed from ESCs” (line 178). Aren’t ESCs PSCs? This resulted confusing for me.

“the diagnosis of AD is defined by the presence of amyloid β and phosphorylated tau protein” (lines 218-219) For most (live) patients, AD diagnosis is not based on these.

“The use of editing techniques can further intensify findings” (line 260) Intensify findings?

iPSCs can “be divided into other cell subtypes from brain tissue” (line 310)  

“It is likely possible to model iPSCs into all somatic cell types” in line 390

“Mutations of A53T mutant α-synuclein or α-synuclein in neurons” (line 337)

“the less invasive acquisition of in vitro study models” line 419

“The modeling of specific diseases by iPSCs has been increasing intensely” (line 420)

“large groups have elucidated points of great danger for these cells(…) applying iPSCs for grafting tend to deeply investigate” lines 434, 435 and 437

These sentences seem incomplete:

“While for epileptogenic cortical malformations or developmental epileptic encephalopathies experimental models using organoids become attractive to generate a natural environment-like culture. (lines 263-265)

Improvements in organoid protocols to enhance reproducibility, specificity, and functional characteristics. (lines 466-467)

Other remarks:

In line 92-93 the authors stated that the use of PBMCs for reprogramming allows “the use of frozen samples stored in blood banks worldwide”, without addressing the ethical issues implied in this statement.

In lines 150-151 the authors indicated that “In summary, identifying molecules that enhance pluripotency and maintain stem cell states is crucial due to the low success rates in current iPSC generation protocols.” This is quite a dramatic statement, as it doesn’t seem to be an issue for many laboratories that regularly obtain iPSCs lines around the world.

Could you please explain the difference between genes “SCNA, LRRK2, PINK1, PARK2” and genes “DJ-1, PARK9 (ATP13A2 ), SJ-1 and VPS35” Not sure what is the distinction between them, suggested in the phrasing of lines 321-322

Please, revise this sentence: “Among the uses, neurons from patients with PD can be used to form organoids to analyze pathogenic mechanisms in detail and test drugs against PD.”

Astrocytes and oligodendrocytes (line 448) are not neuronal cell types

Punctuation in line 340 and 360 include ] and (. Seems like it has not been properly revised.

I would like to make two additional suggestions,

The review revolves around two main topics, iPSCs and organoids. I would suggest a more detailed explanation centered around organoids, since this is a more recent technology, with more relevant developments nowadays.

Also, the extension of the tables seem excessive, while figures are more limited (and figure 2 seems to indicate that human organs can be differentiated from organoids?)

Comments on the Quality of English Language

Comments focused on choice of words were made in the Comments and Suggestions for Authors

Author Response

We are grateful for your comments and guidance to improve our article. We made the necessary corrections that can be seen in the attached file

Reviewer 3 Report

Comments and Suggestions for Authors

As attached, comments of deficiencies concerned paragraphs 2.1 and 3.2 to be in need of additional information and supports.

Author Response

(The authors gave the same response as above.)

Reviewer 4 Report

Comments and Suggestions for Authors

I would like to express my gratitude to the authors and the diligent reviewers for their efforts in evaluating the outstanding papers. However, I find it perplexing that the focus has been exclusively on Alzheimer's disease, epilepsy, Parkinson's disease, and spinal cord injury. A thorough literature search, including PUBMED, indicates that spinal cord injury and cerebrovascular disease are the most prevalent conditions where iPSC-based regenerative therapies hold significant potential. Consequently, the exclusion of cerebrovascular disorders raises questions.

Major points

The overall discussion lacks specificity. For instance, conflicting reports exist on the efficacy of iPSC regenerative therapy for Alzheimer's disease. To provide a more coherent perspective, it would be beneficial to emphasize the majority viewpoint that supports the effectiveness of iPSC regenerative therapies for Alzheimer's disease.

1.       The overall discussion lacks specificity. For instance, conflicting reports exist on the efficacy of iPSC regenerative therapy for Alzheimer's disease. To provide a more coherent perspective, it would be beneficial to emphasize the majority viewpoint that supports the effectiveness of iPSC regenerative therapies for Alzheimer's disease.

2.       The rapid development of iPSC-based therapies for neurological diseases, especially in the context of cerebrovascular diseases, deserves more attention. As highlighted in the general discussion, there is a need for additional reviews on this particular aspect.

3.       It is established that the careful selection of donor cells is a crucial step in applying these new technologies to mitigate potential challenges such as tumor formation. Rather than overemphasizing this point, I suggest presenting a streamlined flow with the essential requirements in order to maintain focus on the broader subject matter.

Author Response

(The authors gave the same response as above.)

Reviewer 5 Report

Comments and Suggestions for Authors

The review overall is good and comprehensive. As acknowledged in the text, although iPSCs have great potential for regenerative therapies, some challenges and limitations need to be addressed before clinical applications. One thing is how to standardize procedures to ensure reliable reproducibility of iPSCs and their effectiveness in the same form of neurodegenerative disease such as AD. As for improvements in organoid protocols, some scaffolding materials/matrices may help to enhance reproducibility and functional characteristics, which should be discussed somehow.

In line 447,  ''key neuronal cell types'' would be better changed to ''key neural cell types''.

Author Response

(The authors gave the same response as above.)

Round 2

Reviewer 1 Report

Comments and Suggestions for Authors

The authors have nicely revised the manuscript. However there are certain concerns which must be addressed. The abstract must reflect the changes that have been made in the manuscript. It would be best to include multiple comprehensive figures possibly for each disease discussed in the manuscript to depict the current state of the art research using IPSCs and Organoids. Please discuss the role of single cell transcriptomics in disease modeling and novel drug discovery using IPSCs and organoid models. Currently the traumatic brain injury section is lacking review of IPS studies such as those done by Ryotaro Imai et al. (2023): Stem Cells 41:6:603-616, Gao, X. et al. (2016) Sci Rep 6:22490; Ramirez, S. et al. (2021): STAR Protocols 2:100987  and various others. Please revise this section thoroughly. Please add a section on spinal cord injury as well.

Comments on the Quality of English Language

There are grammatical mistakes that must be corrected.

Author Response

Dear reviewer

Thank very much for your considerations and you new revision of this manuscript. I would like to apologize and send you new answers here. The new answers are identified as “ANSWER” and highlighted in yellow.

Reviewer 2 Report

Comments and Suggestions for Authors

I am very dissapointed by the reviewed version of the article.

At first glance, it seemed that my review was missed. Upon closer examination, it was realized that the authors didn't consider many suggestions. Of course, I understand that the authors do not need to accept my suggestions or the suggestions of any other reviewer, but the issue that bothers me is that they do not accept the suggestions for the wrong reasons.

I will include some of these wrong reasons, as examples, and insist that the authors revise the previous review before continuing. If the authors or the editors consider that it would be better ignoring my suggestions then please, don't send it back, I will not argue against that decision.

1.1  Methods: The approach to do this review is not clear. The authors do not indicate that the review is systematic, nor explain why some of the works are included or not. Although the expertise of some of the authors support a more freely crafted manuscript, the lack of a systematic approach causes it to cause some deficiencies in certain topics.

R-Your methodology criticism is not desirable for alteration due to not being a review with any statistical analysis or comparison between multiple articles on the field. We focused on using the most relevant articles for each disease, using the most recent studies or all the ones with fewer studies, for therapy, drug screening, and organoid subjects.

Obviously, the authors didn’t understand my suggestions, nor seems to understand what a systematic review is. In the “The PRISMA 2020 statement: An updated guideline for reporting systematic reviews”, starts by indicating the uses of systematic reviews: “Systematic reviews serve many critical roles. They can provide syntheses of the state of knowledge in a field, from which future research priorities can be identified; they can address questions that otherwise could not be answered by individual studies; they can identify problems in primary research that should be rectified in future studies; and they can generate or evaluate theories about how or why phenomena occur. Systematic reviews therefore generate various types of knowledge for different users of reviews (such as patients, healthcare providers, researchers, and policy makers).[1,2]”. This kind of roles seem to fit your goals, but I might be mistaken.

Organization: In some parts the work seems unorganized. Some examples include

2.1  ·  focusing in different diseases in different parts of the manuscript. Although this can be caused by several reasons, it implies that the conclusions might be useful in some parts, but not all

R-As our article focuses on many diseases and their relation to iPSC modeling and usage, it may be seen as chaotic, but the structure is Introduction (iPSC generation), Modeling (Each disease a paragraph), Therapy and drug-screening (A single chapter with a little of each disease difficulty and advantage), and conclusion (Focusing on all difficulties and future perspectives).

The problem I found was not in the different topics treated, what I mean is that while in some sections some diseases are mentioned, in other sections other, different, alternative diseases are mentioned.

2.2  ·    the abbreviation of iPSC is repeated several times along the manuscript

2.3  ·   A big part of the manuscript is focused on hiPSCs and its derivation, and later on organoids, but the approach of both parts is quite different, not dealing with these two topics in a similar, consistent way.

2.4  ·   Huntington’s disease is seldom mentioned until section 3.2.3, and afterwards is not mentioned again? Other diseases are considered in every other part of the review.

Sometimes the sentences lack specificity, and this is problematic for a review.

3.1  ·               In line 28, neurodegenerative diseases “present large extracellular and intracellular factors”. The nature of such factors is not clarified in this paragraph, and no further information is provided, making the significance of this sentence questionable

R- This sentence is related to PD and AD, the paragraphs for each disease explain the factors that have been only written as a mean to introduce the complexity of each disease

I understand that PD and AD are complex diseases, and this sentence is not helping. I checked again the article, and most of the uses for “factors” refers to the transcription factors. In other instances, factors are referred to environmental (Please, note that term is used in different context than extracellular factors!) or genetic/intrinsic factors. Still, it is not clear why the authors add the adjective “large”. I am not saying it is wrong, what I find troubling is its unspecificity, instead of clarifying it results in more confusion.

3.2  ·              Organoids are defined as 3D “in vitro structures of cells under specific conditions.”, but it is not known what the authors mean by “specific conditions”

R  Again the specific conditions are related to each disease for the modeling targeted. In other studies, such as virus or bacteria co-cultures for studying their pathogenesis in the brain, their presence would be the specific conditions.

Again, the problem is the lack of specificity. There are several 3D in vitro structures of cells that are not organoids, and by adding “under specific conditions” do not add any information, so it results in a bad definition of organoids. Either the conditions that differentiate organoids from other 3D cell structures in vitro are indicated or the characteristics that differentiate organoids (origin? Properties? uses?)

5.1   Who can be reprogrammed? (line 63) Is it Who the correct pronoum?

R I believe Who is the right pronoun as it refers to the subject of the sentence.

This is kind of funny. Almost all pronoums can be used as the subject of the sentence, but Who is used for people, and What for things

5.2   Lymphocytes gave rise to iPSCs? (line 86)

R The studies cited following this sentence show as lymphocytes have been programmed to iPSC.

I honestly think the authors did not understood the comments in this section. It is very different “limphocytes gave rise to iPSCs” to “lymphocytes have been programmed to iPSC”.

5.3   Differentiate MEFS into iPSCs? (line 112)

R- Yes, there is a paper that presented an alteration of an MEF to iPSC.

This is the last example that I will coment on in this section. I can understand that some people still don’t understand the difference between “differentiation”, “reprograming” or “induction”, but I honestly don’t understand who can refer to it as “an alteration of an MEF to iPSC” in a scientific context, and is an epitome of the problem I found in several parts of this review.

7.1 In line 92-93 the authors stated that the use of PBMCs for reprogramming allows “the use of frozen samples stored in blood banks worldwide”, without addressing the ethical issues implied in this statement.

R   The ethical regarding regulation and social aspects is not the focus of the article, but if it is of great matter, it can be added

But it has not been added? I don’t know the regulations in your country, but the use of samples stored in blood banks in some countries for reprogramming is very problematic, and including such suggestion without addressing the ethical concerns would be disturbing for people in countries with similar context as the one I work.

7.5 Astrocytes and oligodendrocytes (line 448) are not neuronal cell types

R- The therm used was neural and is defined as “relating to or characteristic of any structure consisting of nerve cells or their processes”. But can be changed to brain cell if preferred.

No, neural is perfect, but neural was not the “therm” used, I checked the original version and the term previously used was neuronal. This sentence has been modified, I don’t understand why the authors try to convince me that neural and not neuronal was written before when I saved a highlighted copy of the manuscript and we all can see the previous versions of it in the webpage.

So I will not include more examples, I honestly think that the corrections previously suggested would improve the article, despite the different answers from the authors.

Comments on the Quality of English Language

In the first review I was not worried by it, after reading the authors response to the review I am not so sure anymore

Author Response

(The authors gave the same response as above.)

Reviewer 3 Report

Comments and Suggestions for Authors

Excellent review and additional information

Author Response

(The authors gave the same response as above.)

Round 3

Reviewer 1 Report

Comments and Suggestions for Authors

The revised manuscript is much better than the original version. However, there are multiple grammatical mistakes which need to be corrected. The quality of figures is not upto the mark. Multiple figures with detailed description must be provided for each disease condition. Without readin the text the reader must be able to understand disease pathology and how organoids and iPSCs are able to provide a tangible solution in terms of therapy, biomarkers or neuropathology. Write a section on direct in vivo reprogramming of astrocytes and microglia into functional neurons in the future directions.

Comments on the Quality of English Language

Manuscript has multiple grammatical errors and must be corrected by a professional native english writer.

Author Response

Dear reviewer

Thank very much for your considerations and you new revision of this manuscript. The new answers are identified as “ANSWER” and highlighted in yellow.

The revised manuscript is much better than the original version. However, there are multiple grammatical mistakes which need to be corrected. The quality of figures is not upto the mark. Multiple figures with detailed description must be provided for each disease condition. Without readin the text the reader must be able to understand disease pathology and how organoids and iPSCs are able to provide a tangible solution in terms of therapy, biomarkers or neuropathology. Write a section on direct in vivo reprogramming of astrocytes and microglia into functional neurons in the future directions.

Answer 1: Thank you for your recommendations. We changed the figure 1 (all diseases mentioned in the text were included)

Answer 2: We wrote a paragraph about future perspectives and presented in vivo reprogramming.

Answer 3: A thorough grammatical review of the text was carried out.

Reviewer 2 Report

Comments and Suggestions for Authors

The authors seem to have addressed all the issues raised. No further changes will be suggested

Comments on the Quality of English Language

No comments

Author Response

Dear reviewer

Thank very much for your considerations and you new revision of this manuscript. The new answers are identified as “ANSWER 2” and highlighted in yellow.

The authors seem to have addressed all the issues raised. No further changes will be suggested.

Answer 3: A thorough grammatical review of the text was carried out.